# Analysis of Reports Sent to the Portuguese Pharmacovigilance System and Published Literature Regarding the Safety of Metformin in the Elderly

**DOI:** 10.3390/healthcare11152197

**Published:** 2023-08-03

**Authors:** Beatriz Esteves, Cristina Monteiro, Ana Paula Coelho Duarte

**Affiliations:** 1Health Science Faculty, University of Beira Interior, 6201-001 Covilhã, Portugal; beatriz.rolao.esteves@ubi.pt; 2UFBI-Pharmacovigilance Unit of Beira Interior, University of Beira Interior, 6201-001 Covilhã, Portugal; apcd@ubi.pt; 3CICS-UBI-Health Sciences Research Centre, University of Beira Interior, 6201-001 Covilhã, Portugal

**Keywords:** adverse drug reactions, metformin, safety, type 2 diabetes mellitus, elderly

## Abstract

The first line medication for the treatment of type 2 diabetes is metformin. This study aims to investigate the safety profile of metformin and metformin combination medications in older adults using pharmacovigilance data. A literature search was used to identify published clinical studies reporting safety of metformin in older patients (age ≥ 65 years old), which were then thoroughly evaluated. Additionally, a deep analysis was performed, taking into account suspected adverse drug reaction (ADR) reports submitted to the Portuguese Pharmacovigilance System involving patients with 65 years old or older, with metformin or metformin combination as the suspected drug. The results suggest that metformin is safer when used in combination with other antidiabetics than when used in monotherapy. Metformin prolonged-release tablets have a lower incidence of adverse effects compared to treatment with immediate-release metformin tablets. The analysis of the reports showed that “gastrointestinal disorders” was one of the most common classes reported, and metformin alone was the drug most commonly associated with serious gastrointestinal reactions that resulted in hospitalization. In addition, it was the drug most commonly associated with the lactic acidosis ADR. Even though most ADRs in the reports were serious, the majority progressed to cure. According to the analysis performed, the results suggest that the patient’s renal function should be considered in order to prevent ADRs associated with metformin, such as lactic acidosis. Therefore, monitoring the safety profile of metformin remains essential to prevent serious ADRs.

## 1. Introduction

The most commonly used medication for type 2 diabetes mellitus (T2DM) in the world is metformin [1]. Despite its relevant efficacy, metformin should be used with caution in older patients because of the high risk of potentially serious and life-threatening adverse effects [2]. The action of metformin on the liver, primarily blocking gluconeogenesis and reducing glycogenolysis and fatty acid oxidation, is the primary mechanism of action for the treatment of diabetes [3]. In addition to this mechanism of action, metformin causes a delay in postprandial absorption and an increase in glucose uptake at the intestinal level [3]. It also increases glucose uptake in muscle and adipose tissue. Non-steroidal anti-inflammatory drugs (NSAIDs), selective cyclooxygenase-2 (COX-2) inhibitors, angiotensin-converting enzyme inhibitors (ACEIs), angiotensin II receptor antagonists (ARA II), and diuretics, particularly loop diuretics, can impair renal function, which may increase the risk of lactic acidosis [4]. Therefore, careful monitoring of renal function is required when these drugs are started or used concomitantly with metformin [4]. Metformin has a minimal risk of causing hypoglycemia, but a higher risk of gastrointestinal side effects and lactic acidosis [2]. According to the Food and Drug Administration (FDA) guidelines, patients with glomerular filtration rates (GFR) less than 30 mL/min are contraindicated for metformin treatment, and metformin should be used with caution in people who have heart failure or impaired liver function, because these conditions may increase the risk of lactic acidosis [5]. Lifestyle modification strategies, such as diet, exercise, and weight management, are crucial in the prevention and treatment of T2DM, in addition to pharmaceutical therapy [2]. Furthermore, renal function decreases with age; therefore, older people receiving metformin should be monitored more closely [5].

However, there are other issues related to the medication which can lead to adverse event in this special population, such as the prescription of an incorrect dosage or the administration of a drug without a clear reason [6]. Before a drug is allowed to be marketed, there is usually a good level of evidence from clinical trials demonstrating its efficacy for the indication and population studied, as well as its safety regarding the most common adverse reactions. However, when a drug begins to be used in clinical practice, some adverse drug reactions (ADRs) are not known, particularly for the populations not included in the clinical trials, as well as those resulting from drug–drug interactions or drug–disease interactions, late-onset ADRs, or chronic exposure. Therefore, it is necessary to continuously monitor, through pharmacovigilance, the safety of marketed medicinal products [7].

It is important to continue monitoring the safety of metformin in the elderly by collecting real-world data from the occurrence of serious ADRs and the effects of concomitant medications. For all these reasons, a review of the literature was performed, considering the safety of metformin in older patients. Afterwards, reports of suspected ADRs reported to the Portuguese Pharmacovigilance System (PPS) between 1999 and 2022 in the elderly were analyzed, with metformin as a suspect drug (used either as a mono- or combination therapy). The overall goal was to draw a conclusion on the safety of metformin in the elderly by taking into account the scientific literature and the actual pharmacovigilance data. Secondarily, we aimed to compare the extent to which the safe use of these drugs in this population is consistent with the drug safety documented in the literature. In fact, to the best of our knowledge, this is the first study which analyzes the outcomes of clinical studies and pharmacovigilance data for this special population.

## 2. Materials and Methods

### 2.1. Comprehensive Review

In order to find papers addressing the safety of metformin and metformin combination drugs in older patients (age ≥ 65), a bibliographic search was performed in two different databases (PubMed and Web of Science,). After a first search with a longer time frame, no relevant articles were found before 2010; therefore, the search included the period between 1 January 2010 and 23 September 2022.This search was carried out using the words: “adverse reaction”, “adverse event”, “safety” or “pharmacovigilance”; “metformin”; “diabetes”; “elderly”, “older people”, “older patient”, “older person”, “geriatric” or “older adult” and “Humans (Mesh)” and filters such as age ≥ 65, articles related with clinical studies, case reports, systematic reviews and meta-analyses, the human species, the English and the Portuguese language were used. In vivo preclinical studies and studies in which drug safety was not described were excluded. Studies with ambiguous design or unclear methods, with non-specific results and those with population younger than 65 years old were also excluded. Finally, studies involving young adults and the elderly whose results were presented as average age, and studies where the results were not separated by specific drugs, were excluded. The results were related to the safety of metformin in the elderly population, a description of associated ADRs, and a conclusion on which drugs are safer in the elderly population.

### 2.2. Analysis of ADRs Reports Sent to the Portuguese Pharmacovigilance System

The suspected ADRs submitted to the PPS were observed and retrospectively examined. The National Authority of Medicines and Health Products, I.P (INFARMED), is responsible for coordinating the PPS. Only reports mentioning metformin and/or metformin combination drugs as suspected drug(s) in patients 65 years of age or older were considered in this study, which covered the years 1999 to 2022. Duplicate reports and reports that did not mention age were among the reports that did not include the data required to classify ADRs. It is also important to note that although each report relates to a singular individual, it is possible that more than one implicated drug and more than one suspected ADR are associated with it. Initially, 485 reports were taken into consideration, of which 119 were duplicate and 16 were excluded. Only 350 spontaneous reports involving individuals who are 65 years old or older were therefore included in the analysis. 

The study variables are represented in Figure 1.

The MedDRA SOC was used to categorize the suspected ADR complaints. MedDRA corresponds to the medical dictionary for regulatory activities and was developed by the International Council for Harmonisation of Technical Requirements for Pharmaceuticals for Human Use to facilitate sharing of regulatory information internationally for medical products used by humans [8]. A more thorough analysis of SOC “metabolic and nutritional disorders”, “gastrointestinal disorders” and “cardiac disorders” was conducted, carefully examining ADRs that resulted in hospitalization using the PT reactions of the MedDRA dictionary. A deeper study of each ADR was also carried out in the reports that had fatal outcomes. The criteria used by the PPS and the World Health Organization-Uppsala Monitoring Center (WHO-UMC) system for determining case causality were observed when determining the relationship between exposure and death. This approach categorizes the causality as certain, probable, possible, unlikely, conditional, or unclassifiable, based on the clinical and pharmacological aspects of the reported history and the quality of the documentation provided [9]. The ADR reports were divided into two categories based on their seriousness: serious and not serious. According to the Guidelines on Pharmacovigilance for Medicinal Products for Human Use, a serious ADR is one that results in death or is life-threatening, keeps the patient in the hospital for a long period of time, leaves them permanently disabled, or results in birth defect(s) [10].

Information on the gender and age of individuals affected by ADRs was analyzed. The patients were divided into the following age categories among the elderly in the study: ages 65 to 74 years, 75 to 84 years, and 85 years and older. Although all reports involved people aged 65 or older, there were reports in which the exact age of the patient was not specified; in these cases, age was categorized as unknown. Gender was categorized as male or female, but similar to age, there were reports in which the patient’s gender was not specified, and these were classified as unknown.

Even though the SOC group “cardiac disorders” is not one of the SOC groups with the highest prevalence, cardiac disorders were frequently mentioned in the articles, so an analysis of this group was also performed.

The data were organized according to the variables studied, and statistical analysis was performed using a descriptive analysis processed through the Microsoft Office Excel 365 tool.

## 3. Results

### 3.1. Comprehensive Review

In the review performed, 13 articles met the criteria stated in the methods for our analysis. The study years of the selected subjects were 2021 (n = 1), 2019 (n = 1), 2018 (n = 1), 2017 (n = 2), 2016 (n = 1), 2014 (n = 1), 2013 (n = 2), 2012 (n = 1), 2010 (n = 3). Of the 13 articles included in the review, 7 were observational studies, 3 were clinical trials, 1 was a systematic review and 2 were case reports.

Main Ideas Supported by the Literature Review

Table 1 provides a summary analysis of the studies gathered using the search technique used in this in-depth review. In summary, the studies showed that patients taking metformin extended-release tablets had a lower incidence of adverse effects compared to the treatment with metformin immediate-release tablets [11]. Regarding gastrointestinal adverse effects and hypoglycemia, the studies showed that metformin in combination with voglibose had a lower incidence of these effects than metformin monotherapy. In addition, individuals experienced a greater weight loss after treatment with this combination compared to metformin monotherapy [12]. Concerning lactic acidosis, in the study by Hooda et al., the patient suffered dehydration and subsequently developed acute renal injury after taking liraglutide, and because he was also taking metformin, there was an accumulation of metformin, leading to the development of lactic acidosis [13]. 

The combination of metformin with gemigliptin was found to be more effective than monotherapy with either drug, without safety concerns [14]. According to a systematic review by Schlender et al., the safety and efficacy profiles of metformin have been shown to be better than other treatments for the control of T2DM in the elderly [15]. According to Becquemont et al., after a 3-year follow-up, there was no increase in mortality in the 25% of elderly patients medicated with metformin who received a dose not adjusted for renal function [16]. The study by Margiani et al. found that metformin accumulation can lead to the development of lactic acidosis, in this case as a result of pre-renal injury brought on by ileostomy [17]. When it comes to the adverse cardiovascular events, patients taking metformin or glimepiride had a lower risk of nonfatal cardiovascular events than those taking gliburide [18]. Initial treatment of T2DM with sulphonylureas was associated with a higher risk of cardiovascular events and death than with metformin [19]. In the study by Moore et al., metformin was associated with poorer cognitive outcomes, possibly related to the high risk of vitamin B_12_ deficiency. However, vitamin B_12_ and calcium supplements may treat vitamin B_12_ deficiency caused by metformin, and contribute to improved cognitive outcomes [20]. Regarding mortality, studies have also found that metformin use can decrease mortality when used as secondary prevention [21], and that mortality is lower in patients taking metformin as monotherapy or in combination, compared to those not taking antidiabetic drugs [22]. There were also fewer deaths in patients taking metformin in monotherapy or in combination with sulphonylureas, compared to sulfonylurea therapy alone [23].

**Table 1 healthcare-11-02197-t001:** Studies evaluating the safety of metformin and the metformin combination in the elderly.

References	Type of Study and Duration of the Study	Study Population	Number of Patients Aged ≥ 65 Years	Number of Patients Aged < 65 Years	Drugs Compared/Route of Administration	Outcomes	Study Limitations
Guo et al., 2021 [11]	Randomized, open and parallel controlled clinical trial. -2 years and 7 months	Patients with T2DM and taking metformin.	n = 150	n = 736	Prolonged-release metformin tablets, immediate-release metformin tablets; oral	-Long-release metformin tablets and immediate-release metformin tablets; similar therapeutic efficacy in the treatment of T2DM. -Long-release tablets: lower incidence of adverse effects.	-Questionnaire used does not have sufficient sensitivity to assess the quality of life of users. -Failure to obtain circulating GLP-1 levels of 60 blood samples.
Oh et al., 2019 [12]	Multicenter, randomized, double-blind, and parallel group study. -24 weeks	Patients with T2DM and inadequate glycemic control	n = 38	n = 149	Metformin alone and metformin in combination with voglibose (vogmet); oral	-Adverse effects hypoglycemia: minors in the vogmet-treated group compared to the metformin-treated group alone. -More significant weight loss when using vogmet treatment.	-Absence of a group alone with voglibose.
Hooda et al., 2018 [13]	Case report	A 70-year-old man with T2DM, without microvascular or macrovascular complications, with class 2 obesity, hypertension, dyslipidemia, and hypothyroidism	n = 1	n = 0	Metformin and liraglutide; oral and subcutaneous.	-Dehydration and development of acute kidney injury after the use of liraglutide (the main side effects of which are vomiting and nausea). -Development of lactic acidosis associated with metformin.	-Rapid titration of the liraglutide dose without adequate follow-up.
Lim et al., 2017 [14]	Multinational, multicenter, randomized, active-controlled, double-blind, phase III trial. -1 year and 8 months	Patients with T2DM who do not take any antidiabetic	n = 77	n = 356	Metformin in combination with gemigliptin, metformin monotherapy or gemigliptin; oral.	-Metformin in combination with gemigliptin: efficacy superior to monotherapy with each drug.	-Reduced number of patients using metformin alone. -No safety concerns.
Schlender at al., 2017 [15]	Systematic review	Elderly individuals aged ≥65 with T2DM	n = 230229	n = 0	Metformin (alone or in combination), placebo or other antidiabetics (gliburide, glimepiride, thiazolidinediones, tolbutamide, biguanides, vildagliptin, rosiglitazone and pioglitazone); oral.	-Safety and efficacy profiles of metformin appear to be better than those of other treatments for the control of T2DM in the elderly.	-Reduced quality and amount of evidence due to lack of information related to adverse effects, including gastrointestinal changes or kidney failure.
Becquemont et al., 2016 [16]	Prospective cohort study -3 years	Non-institutionalized patients aged ≥ 65 years and with chronic pain, T2DM or AF.	n = 3434	n = 0	Metformin or digoxin or spironolactone; oral.	-Approximately 25% of patients taking metformin receive a dose that is not adapted to renal function, but there is no increase in mortality after a 3-year follow-up. -Renal failure increases the risk of developing lactic acidosis associated with metformin therapy, however none of the deaths were associated with lactic acidosis.	-Small number of patients receiving metformin. -Risk of residual confounding factors and lack of diversity of the group of patients included in this study.
Margiani et al., 2014 [17]	Case report	A 70-year-old man with T2DM, prostate hypertrophy, hypertension and who was subjected to temporary ileostomy.	n = 1	n = 0	Metformin; oral.	-After ileostomy, dehydration and electrolyte imbalances were observed. -These changes led to the development of a pre-kidney injury. -Consequently, there was an accumulation of metformin which resulted in serious lactic acidosis.	-The patient was also undergoing treatment with diuretics which contributed to the worsening of dehydration and electrolyte imbalances resulting from ileostomy.
Hung, 2013 [18]	Retrospective cohort study based on the study population.	Patients with T2DM, with no history of cardiovascular disease and aged ≥ 30 years.	n = 231	n = 928	Monotherapy with metformin, glimepiride or gliburide; oral	-Lower risk of developing non-fatal cardiovascular events in the group taking metformin or glimepiride compared to the gliburide therapy group.	-Insufficient information regarding patient comorbidities.
Moore et al., 2013 [20]	Cross-sectional study	Patients with T2DM and Alzheimer’s or moderate cognitive dysfunction or individuals with intact cognitive function.	n = 1164	n = 190	Metformin alone and metformin in combination with calcium and vitamin B_12_ supplements; oral	-Use of metformin associated with altered cognitive performance. -Vitamin B_12_ and calcium supplements may improve metformin-induced vitamin B_12_ deficiency and contribute to better cognitive results.	-Insufficient information regarding the duration of metformin treatment, severity of diabetes and the use of other antidiabetics. -Reduced sample.
Roumie, 2012 [19]	Retrospective cohort study	Patients with T2DM and AMI or stroke	n = 118,014	n = 135,626	Monotherapy with metformin or sulphonylureas; oral	-Initial treatment of T2DM with sulphonylureas associated with a higher risk of cardiovascular events and death than with metformin.	-Inaccuracies in measurements associated with results not coming from the central laboratory.
Roussel et al., 2010 [21]	-Prospective observational study -2 years	Patients with T2DM and atherothrombosis	n = 12,649	n = 6.904	Treatment with or without metformin; oral	-Use of metformin as secondary prevention may decrease mortality.	-Ignorance of the duration of diabetes and metformin use. -Lack of information on the level of glycated hemoglobin.
MacDonald, 2010 [22]	Case-control study	Patients with T2DM and HF	n = 3102	n = 164	Monotherapy with metformin, metformin in combination; oral	-Lower mortality in patients taking metformin alone or in combination with users who were not taking antidiabetics.	-Study was based on medical diagnoses and existing documents of heart failure, comorbidity and risk factors. -There was no independent confirmation of the diagnoses.
Evans, 2010 [23]	Population-based prospective cohort study	Patients with T2DM and CHF	n = 365	n = 57	Monotherapy with metformin or sulphonylureas and association of both drugs; oral	-Fewer deaths in patients taking metformin alone or in combination with sulphonylureas compared to sulphonylureas therapy alone.	-Confounding variables that may have contributed to the creation of bias in the differences observed between the groups considered.

T2DM, type 2 diabetes mellitus; AF, atrial fibrillation; HF, heart failure; CHF, chronic heart failure; GLP-1, glucagon-like peptide-1; AMI, acute myocardial infarction.

### 3.2. Portuguese Pharmacovigilance System-Adverse Drug Reaction Analysis

During the period studied, a total of 350 ADRs reports were analyzed, and it has been observed that there have generally been more reports over time.

In the analysis of the type of person who reported the ADR, it was concluded that most of the reports were sent by the marketing authorization holders (163 reports, which corresponds to 46%), followed by pharmacists and physicians with 26% (91 reports) and 24% (85 reports), respectively. Users or other non-healthcare professionals represented 3% of the people who reported (9 reports), and other health professionals accounted for a percentage of 1% (4 reports).

The majority of the ADRs occurred in the age range of 65 to 74 years (190 reports), with females most commonly affected (107 reports). In each age group, the number of reports was found to be greater in the female gender than in the male gender, with the number of reports generally declining with age. The existence of reports without gender mentioned occurred for all age groups, with the exception of individuals who were 85 years or older. Of the 21 notifications without defined age, 8 were female, 5 male and 8 of unknown gender.

A deeper analysis showed that different SOCs are affected depending on whether metformin is used alone, in a fixed combination, or in combination with other classes of drugs (Table 2).

Overall, in SOC group, “Metabolism and nutrition disorders” were the most reported, followed by “Gastrointestinal disorders” and “General disorders and administration site conditions” (Table 2).

All the Preferred Term (PT) reactions were analyzed, and according to Table 3, PT reactions differ depending on whether metformin is used alone, in a fixed combination, or in combination with other classes of drugs.

In general, lactic acidosis, diarrhea, hypoglycemia, and vomiting were the most commonly reported PT reactions (Table 3).

Based on the SmPC of the suspected drugs and a number of scientific papers, it was verified whether the ADR had been previously described or not.

The 350 ADR reports analyzed included a total of 1261 reactions; of these, 869 reactions are described, 358 reactions are not described, and 34 reactions are categorized as not applicable (described as drug ineffective).

According to the ADRs’ seriousness, it was observed that the majority of the ADRs, 68% (or 237 reports), were serious, while the remaining 32% (or 113 reports) were non-serious.

Hospitalization was the most reported seriousness criterion in ADR notifications, with a percentage of 36% (84 reports); this was followed by life risk, with a percentage of 27% (65 reports), clinically important criteria, with a percentage of 26% (61 reports), and death and disability, with a percentage of 7% (17 reports) and 4% (10 reports), respectively.

A deeper analysis of the ADRs from the IME list was also performed (Table 4). Metformin was the target drug most reported. 

A characterization of ADRs with terms of MedDRA terminology belonging to the designated medical event (DME) list was also performed (Table 5), and metformin was also associated with several ADRs.

A deeper analysis of the reports with a fatal outcome showed that 47% of them (i.e., 8 out of a total of 17 deaths) were attributed to metformin, a suspected drug. This was followed by metformin and other drugs, especially glibenclamide, chlorothalide, ramipril, clopidrogel, atorvastatin, vildagliptin, alprazolam, pantoprazole, acarbose, gliclazide and empaglifozin, and metformin in fixed combination with dapagliflozin.

The causal relationship between those outcomes and the suspected drugs, where the seriousness criterion was death, was probable in two of the deaths, possible in three of the deaths, conditional on two of the deaths, and not established in the remaining ten deaths.

Most ADRs with the outcome death occurred in the age group 65–74 years, with 12 deaths, mostly male (11 deaths), followed by the age group 75–84 years, with 3 cases, all of which were female. The age group of 85 years and older included two cases, one female and one male.

The number of hospitalizations for metformin as the suspected responsible drug was equal to the number of hospitalizations for metformin and other medications, i.e., 27 hospitalizations, which corresponds to 32%. The number of hospitalizations for metformin in fixed combination was 30, which corresponds to 36% as a percentage.

Most of the reports wherein hospitalization occurred (32 hospitalizations) were associated with the age group 75–84 years, and mostly involved males (18 hospitalizations); this was followed by the 65–74 age group, with 30 hospitalizations (14 females, 15 males, and 1 unknown). For individuals aged 85 years or older, 13 hospitalizations (11 in females 2 in males) occurred. 

The SOC group “cardiac disorders” was subjected to a more thorough analysis, as explained before. Table 6 shows the reported serious ADRs that required hospitalization.

Relative to the SOC group, “metabolism and nutrition disorders” (one of the most SOC reported), a more detailed analysis (Table 7) was performed. In this SOC, 80.1% (n = 149) of the reports were serious, 60 were life-threatening, 56 resulted in hospitalization, 2 resulted in disability, 17 were clinically important, and 14 resulted in death. Metformin alone was the drug most associated with serious reports, and the most prevalent reaction in this group was lactic acidosis, with 25 occurrences.

Concerning the SOC “gastrointestinal disorders”, 56.2% (n = 73) of the reports were serious; 14 were life-threatening, 22 resulted in hospitalization, 9 resulted in disability, and 28 were clinically important. In the reports with the outcome hospitalization, metformin was the drug mostly reported (Table 8). 

Considering the reports with a fatal outcome, 28 reports were associated with the use of metformin in monotherapy as a suspect drug. In the remaining notifications, metformin was associated with other drugs (Table 9).

After the evaluation of the reports according to the criterion adopted by the PPS and the WHO-UMC causality assessment system, as described in the method section, metabolic acidosis and lactic acidosis were considered likely related to the use of metformin alone. Additionally, renal failure, decreased blood pH, shock, respiratory failure, and fatigue were considered to be probably related to the use of metformin alone. The remaining reactions described in Table 9 associated with metformin alone showed a possible, conditional, and unknown causal relationship. Lactic acidosis was considered possibly related to the use of metformin + dapagliflozin in a fixed association. For the remaining reports, no causal relationship has been established.

Finally, an assessment of the patient’s clinical status was performed. Most patients progressed to cure (43%, 151 reports), and 4 to cure with sequelae (1%); 27 were in recovery (8%), 15 continued without recovery (4%), and 17 evolved to death, representing 5% of the total reports. However, in 136 of the reports (39%), the evolution result was unknown.

## 4. Discussion

To determine the safety of metformin in elderly patients, information from the literature review and the ADRs reports sent to the PPS were analyzed. 

Metformin extended-release tablets have a lower incidence of adverse effects compared to treatment with immediate-release metformin tablets [11]. Studies have shown that metformin in combination with voglibose, an association not available in Portugal, showed a lower incidence of gastrointestinal effects than metformin alone [12]. 

In general, there has been a discontinuation of metformin use due to gastrointestinal adverse effects [24]. In fact, the analysis of the reports showed that “gastrointestinal disorders” was one of the most reported SOC, and metformin alone was the drug most commonly associated with serious reports of gastrointestinal reactions that resulted in hospitalization. These results are compatible with the known safety profile for these medicines [4].

Concerning the lactic acidosis cases, according to the literature, metformin therapy is not always adapted to renal function [16]. Lactic acidosis is a serious complication that can result from the metformin administration and is associated with significant morbidity and mortality. There is a higher risk of developing lactic acidosis when renal function is impaired [13,17,25]. Effectively, the analysis of ADRs with a fatal outcome showed that lactic acidosis was considered probably related to the use of metformin alone. In addition, in the analysis of serious reports of reactions of the SOC “metabolism and nutrition disorders” that resulted in hospitalization, it was found that metformin alone was the drug most commonly associated with these reactions. According to Becquemont, when an individual has renal failure, metformin therapy may contribute to an increased risk of developing lactic acidosis [16]. Additionally, changes in renal function are common in the elderly, and may affect drug elimination, which explains the hypoglycemia. In fact, hypoglycemia remains a critical concern in elderly patients with diabetes, because in addition to renal impairment, these patients can have others predisposing factors, such as cognitive impairment, that can affect glycemic targets [13,15]. However, since the renal function of each of these patients was unknown, these results are not conclusive. 

A study by Moore et al. concluded that metformin use was associated with a change in cognitive performance caused by vitamin B_12_ deficiency associated with metformin therapy [20]. According to the analysis performed, there were four occurrences of vitamin B_12_ deficiency in the SOC “Metabolism and nutrition disorders”. 

Additionally, the analysis of the literature showed that the initial treatment of T2DM with sulphonylureas was associated with a higher risk of cardiovascular events and death than with metformin [19]. In the analysis of suspected drugs associated with serious reports of cardiac reactions resulting in hospitalization, sulphonylurea (glibenclamide) was present in only one occurrence in combination with metformin.

Considering the reports of the PPS, according to INFARMED, the number of reports has been increasing [26,27]. This growth is reflected in the results obtained, since the total number of reports has increased over the years. 

In general, the physicians and the marketing authorization holders were the largest reporters, followed by the pharmacists [27]. Nurses, health institutions, other health professionals and users reported less to the PPS [27]. The results obtained are in accordance with these data.

Regarding the demographic data obtained and considering that the aging contributes to the development of chronic diseases and to a greater susceptibility to the development of ADRs, it is observed that the most ADRs occurred in the age group 65–74 years, and that the number of reports generally decreased with increasing age. These results are due to the fact that the population in this age group is disproportionately represented compared to the other age groups [28,29,30]. 

In each age group, the number of reports is higher for females compared to males, which may be related to the fact that women are more prone to adverse effects than men due to a combination of pharmacokinetic and pharmacodynamic factors [31]. However, there are reports wherein gender and age are unknown.

Considering the number of reports analyzed (350), the total number of reactions was 1261, of which 869 were described in the summary of drug characteristics or in scientific articles, 358 were not described, and 34 reactions were classified as not applicable. The reactions considered as inapplicable were related to those that appeared described as drug ineffective, a term that has not been described in the summary of drug characteristics, which may be related to the lack of efficacy of the drug, which can be associated with different causes such as inappropriate use of medication and factors related to interindividual variability [32].

Of the 350 notifications, 237 were serious, and the most prevalent seriousness criterion in most cases was hospitalization, which demonstrates the seriousness of the ADRs that users developed; however, generally, individuals and health professionals have a greater sensitivity and tendency to report serious ADRs. This may also explain the results obtained [33,34]. 

Metformin alone is associated with several ADRs with terms of MedDRA terminology belonging to the IME and DME list, but as mentioned earlier, metformin is the most commonly prescribed antidiabetic drug worldwide [1]. Most ADRs with a fatal outcome were associated with the age group from 65 to 74 years and to the male gender, which may be related to the fact that the elderly population tends to have a higher demographic distribution in this age group compared with the other age groups studied [30]. In most cases, no causal relationship was established between ADRs and drugs, so these deaths may not have been related to metformin. This highlights the importance of obtaining high-quality reports [35].

For ADRs wherein the seriousness criterion was hospitalization, the prevalence of hospitalizations was higher in the case of metformin in a fixed association, but it was very close to the prevalence associated with metformin and metformin and other medications, which presented the same number of hospitalizations. Most ADRs corresponded to the age group of 75 to 84 years, and when it comes to gender, these were mostly reported in male patients. This may be due to the fact that individuals in this age group are physically debilitated, i.e., the elderly have more comorbidities as they age, are prone to be polymedicated, and also show more pronounced pharmacokinetic and pharmacodynamic changes (with greater susceptibility to developing ADRs) [29,36].

However, most cases evolved favorably towards recovery.

Despite some limitations, this study allowed a review of the safety of metformin in the elderly. The major challenge was the variety of different studies selected for the review, which made comparison difficult. The analyzed data were obtained from observational studies and clinical trials in which patients were selected through various inclusion and exclusion criteria. Numerous doses of metformin were studied, and not all studies included a placebo group. Each trial’s findings were measured differently, which could have influenced the outcomes.

Additionally, data from PPS must be carefully interpreted because a fatal outcome does not always indicate a causal relationship between the suspect drug and the outcome. It is also important to mention the difficulty of determining a causal relationship between a few ADRs and suspected drugs, because of information gaps in the reports. In addition, healthcare professionals in general do not report already-known ADRs for a determined drug, which can explain why metformin has received so few reports in PPS [37].

## 5. Conclusions

Metformin is indicated as a first-line treatment for T2DM, but some ADRs have been ascribed to and should be considered in its usage. To avoid metformin-associated ADRs such as lactic acidosis, patients’ renal function should be considered. All ADRs should continue to be reported for the protection of users and public health. In the future, it is important that further work in the area of pharmacovigilance is performed to investigate whether lactic acidosis could constitute a very frequent ADR of metformin, even with intact renal function. It is also important that healthcare professionals as well as users continue reporting, while ensuring these reports provide detailed clinical information at the time of reporting. Thus, working towards the safe use of metformin, it is necessary to implement continuous and adequate monitoring of this type of therapy in the elderly. 

## Figures and Tables

**Figure 1 healthcare-11-02197-f001:**
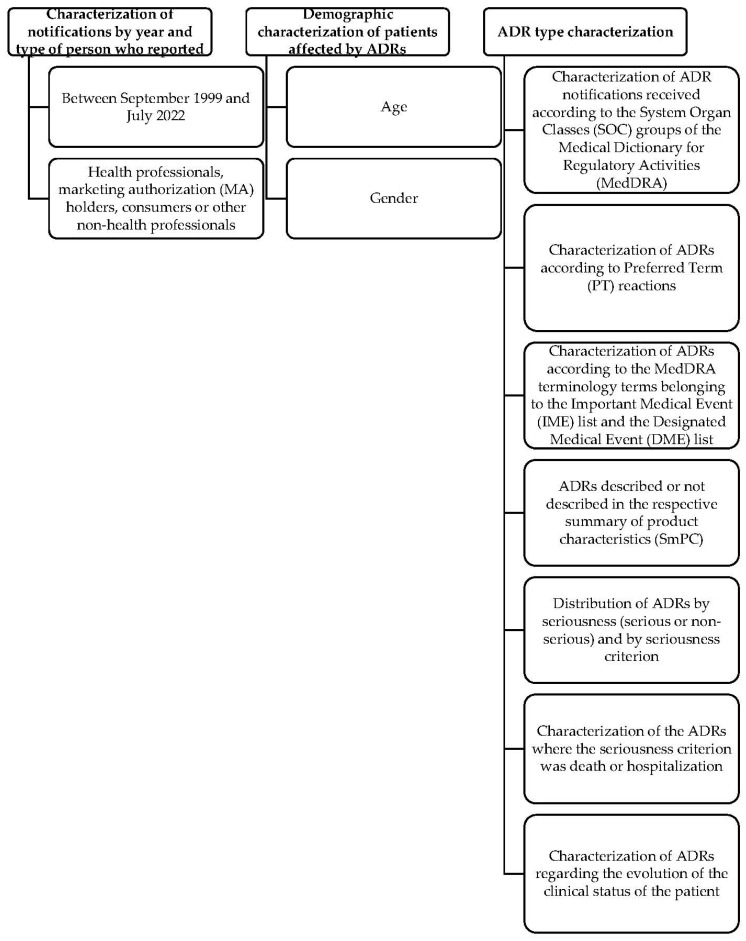
Characterization of the study variables.

**Table 2 healthcare-11-02197-t002:** Characterization of adverse drug reactions by SOC affected.

System Organ Classes (SOC)	Metformin	Metformin in Fixed Combination	Metformin and Other Drugs
Blood and lymphatic system disorders	6	1	7
Cardiac disorders	7	10	8
Musculoskeletal and connective tissue disorders	1	10	2
Eye disorders	0	4	0
Gastrointestinal disorders	54	59	17
General disorders and administration site conditions	28	28	18
Nervous system disorders	20	18	14
Metabolism and nutrition disorders	82	57	47
Skin and subcutaneous tissue disorders	13	18	8
Respiratory, thoracic, and mediastinal diseases	4	15	4
Psychiatric disorders	2	8	3
Infections and infestations	1	7	5
Investigations	13	21	17
Immune system disorders	0	0	3
Neoplasms benign, malignant, and unspecified (including cysts and polyps)	1	3	1
Renal and urinary disorders	20	23	14
Vascular disorders	13	3	7
Injury, poisoning, and procedural complications	10	12	6
Hepatobiliary disorders	0	0	6
Congenital, familial, and genetic disorders	0	0	1
Surgical and medical procedures	0	2	1
Reproductive system and breast disorders	0	8	0

**Table 3 healthcare-11-02197-t003:** The most representative adverse drug reactions according to PT reactions.

PT Reaction	Metformin	Metformin in Combination	Metformin and Other Drugs
Diarrhea	29	24	4
Nausea	8	12	7
Abdominal pain	13	6	3
Vomiting	14	17	7
Acute renal injury	12	6	8
Hyperlactacidemia	8	7	0
Lactic acidosis	54	22	5
Metabolic acidosis	14	3	7
Diabetic ketoacidosis/Euglycemic diabetic ketoacidosis	5	4	7
Hypoglycemia	12	10	19

**Table 4 healthcare-11-02197-t004:** Characterization of adverse drug reactions with terms of MedDRA terminology belonging to the list of important medical events (IME) (n being the number of occurrences).

Terms of the IME List (n)	Suspected Drugs (n) *
Inflammatory myofibroblastic tumor (1)	Metformin (1)
Hemolytic anemia (1)	Metformin (1)
Lactic acidosis (71)	Metformin (52); Metformin + Vildagliptin, Acemetacin (2); Metformin + Vildagliptin (5); Metformin + Vildagliptin, Metformin (2); Metformin + Sitagliptin (6); Metformin, Furosemide, Metformin + Sitagliptin (1); Metformin + Sitagliptin, Metformin (1); Metformin + Alogliptin, Metformin (1); Naproxen, Metformin + Vildagliptin (1)
Hyperkalemia (5)	Metformin (3); Metformin, Sitagliptin, Dapagliflozin, Human insulin (1); Etoricoxib, Gliclazide, Lisinopril, Metformin, Naproxen, Paracetamol, Pravastatin, Sodium Risedronate (1)
Sweat gland tumor (1)	Metformin, Ramipril, Tamsulosin (1)
Pulmonary hypertension (1)	Indapamide, Metformin, Sertraline, Losartan, Simvastatin (1)
Acute renal injury (14)	Metformin (8); Metformin + Vildagliptin, Metformin (1); Metformin + Sitagliptin (2); Naproxen, Metformin + Vildagliptin (1); Etoricoxib, Gliclazide, Lisinopril, Metformin, Naproxen, Paracetamol, Pravastatin, Sodium Risedronate (1); Indapamide, Metformin, Amlodipine + Telmisartan (1)
Apnea (1)	Indapamide, Metformin, Sertraline, Losartan, Simvastatin (1)
Cardiorespiratory arrest (3)	Metformin (3)
Tubulointerstitial nephritis (1)	Indapamide, Metformin, Amlodipine + Telmisartan (1)
Rheumatic polymyalgia (1)	Etoricoxib, Gliclazide, Lisinopril, Metformin, Naproxen, Paracetamol, Pravastatin, Sodium Risedronate (1)
Altered state of consciousness (7)	Metformin (5); Metformin + Sitagliptin (1); Glibenclamide + Metformin (1)
Hemiparesis (1)	Metformin (1)
Pulmonary embolism (1)	Metformin + Vildagliptin, Metformin (1)
Anuria (2)	Metformin (1); Metformin + Sitagliptin (1)
Renal failure (4)	Metformin (3); Ranitidine, Metformin (1)
Hematochezia (1)	Metformin (1)
Cardiac arrest (2)	Metformin, Furosemide, Metformin + Sitagliptin (2)
Prerenal insufficiency (1)	Metformin (1)
Renal injury (4)	Metformin (2); Metformin + Sitagliptin (1); Metformin, Ibuprofen (1)
Lupus-like syndrome (1)	Metformin + Sitagliptin (1)
Multiform Erythema (1)	Metformin (1)
Respiratory failure (2)	Metformin (1); Metformin + Vildagliptin (1)
Ischemic stroke (1)	Lisinopril + Hydrochlorothiazide, Metformin (1)
Inadequate diabetes control (5)	Metformin (2); Prednisolone, Metformin (1); Metformin, Vildagliptin, Gliclazide (1); Metformin, Sertraline (1)
Acute pancreatitis (1)	Terbinafine, Metformin, Prednisolone, Vildagliptin, Prednisolone, Metformin (1)
Cholangitis (1)	Metformin, Perindopril (1)
Cholestasis (1)	Metformin, Perindopril (1)
Bacteriemia (1)	Prednisolone, Metformin (1)
Autoimmune hepatitis (1)	Metformin, Perindopril (1)
Duodenal ulcer (1)	Clopidogrel, Metformin, Nebivolol, Telmisartan + Hydrochlorothiazide, Amlodipine, Amlodipine, Atorvastatin, Furosemide, Pantoprazole (1)
Hypoglycemic encephalopathy (1)	Metformin, Glibenclamide (1)
Metastatic pancreatic carcinoma (1)	Dry extract of unfermented leaves of *Camellia sinensis*, Simvastatin, Amlodipine + Valsartan; Bisoprolol, Metformin + Vildagliptin (1)
Renal ischemia (1)	Clopidogrel, Metformin, Nebivolol; Telmisartan + Hydrochlorothiazide, Amlodipine, Amlodipine, Atorvastatin, Furosemide, Pantoprazole (1)
Stroke (1)	Tenecteplase, Metformin (1)
Atrial fibrillation (1)	Tenecteplase, Metformin (1)
Basal cell carcinoma (1)	Acetylsalicylic acid, Metformin + Vildagliptin, Perindopril + Indapamide, Tamsulosin, Rosuvastatin (1)
Toxic skin rash (4)	Fluoxetine, Metformin (1); Prednisolone, Metformin (1); Terbinafine, Metformin, Prednisolone, Vildagliptin, Prednisolone, Metformin (1); Metformin + Sitagliptin, Amlodipine, Hydroxyzine, Acetylsalicylic Acid, Valsartan + Hydrochlorothiazide (1)
Metabolic decompensation of diabetes (1)	Terbinafine, Metformin, Prednisolone, Vildagliptin, Prednisolone, Metformin (1)
Hypoglycemic coma (1)	Glibenclamide + Metformin (1)
Microscopic colitis (1)	Calcitriol, Carvedilol, Clopidogrel, Diazepam, Esomeprazole, Furosemide, Metformin, Levothyroxine Sodium, Trazodone (1)
Craniocerebral injury (1)	Metformin (1)
Erectile dysfunction (1)	Metformin + Sitagliptin (1)
Acute cholecystitis (1)	Metformin; Vildagliptin, Gliclazide (1)
Multiple organ dysfunction syndrome (1)	Metformin (1)
Autoimmune disorder (1)	Metformin, Perindopril (1)
Syncope (1)	Glimepiride, Glibenclamide + Metformin, Human Insulin (1)
Myocardial infarction (1)	Metformin + Sitagliptin (1)
Euglycemic diabetic ketoacidosis (2)	Metformin (2)
Parkinson’s disease (1)	Metformin + Sitagliptin (1)
Pemphigoid (3)	Metformin (1); Metformin + Vildagliptin (1); Metformin + Vildagliptin, Irbesartan + Hydrochlorothiazide, Gliclazide, Dutasteride (1)
Coma (1)	Metformin, Ibuprofen (1)
Upper gastrointestinal bleeding (1)	Metformin (1)
Chronic renal disease (1)	Hydrochlorothiazide + Amyloid, Metformin + Sitagliptin (1)
Brash syndrome (1)	Bisoprolol, Metformin (1)
Necrotizing esophagitis (2)	Metformin (2)
Diabetic ketoacidosis (4)	Metformin (3); Metformin + Sitagliptin (1)
Metabolic acidosis (1)	Lisinopril + Amlodipine, Bisoprolol; Metformin + Vildagliptin; Ramipril (1)
Troponin I increased (1)	Lisinopril + Amlodipine, Bisoprolol, Metformin + Vildagliptin; Ramipril (1)
Cerebral artery occlusion (1)	Indapamide, Metformin, Sertraline, Losartan, Simvastatin (1)
Aortic thrombosis (1)	Metformin (1)
Renal artery thrombosis (1)	Clopidogrel, Metformin, Nebivolol, Telmisartan + Hydrochlorothiazide, Amlodipine, Amlodipine, Atorvastatin, Furosemide, Pantoprazole (1)
Bradycardia (4)	Metformin (1); Glibenclamide + Metformin (1); Lisinopril + Amlodipine, Bisoprolol, Metformin + Vildagliptin, Ramipril (1); Indapamide, Metformin, Sertraline, Losartan, Simvastatin (1)
Myelopathy (1)	Metformin (1)
Hematemesis (1)	Clopidogrel, Metformin, Nebivolol, Telmisartan + Hydrochlorothiazide, Amlodipine, Amlodipine, Atorvastatin, Furosemide, Pantoprazole (1)
Hypothermia (2)	Metformin (2)
Shock (4)	Metformin (3); Metformin + Vildagliptin, Metformin (1)
Bullous dermatitis (1)	Metformin + Vildagliptin (1)
Heart failure (1)	Metformin (1)
Death (1)	Metformin (1)

* In suspected drugs, each n corresponds to the drug or drugs suspected of causing the described reaction.

**Table 5 healthcare-11-02197-t005:** Characterization of adverse drug reactions with terms of MedDRA terminology belonging to the designated medical event (DME) list (with n being the number of occurrences).

Terms of the DME List (n)	Drugs (n) *
Hemolytic anemia (1) Acute kidney injury (9) Renal failure (3) Multiform erythema (1)	Metformin (14)
Autoimmune hepatitis (1)	Metformin; Perindopril (1)
Autoimmune pancreatitis (2)	Metformin; Sitagliptin (1) Metformin; Gliclazide (1)
Acute renal injury (10)	Metformin + Vildagliptin (1) Naproxen; Metformin + Vildagliptin (1) Etoricoxib; Gliclazide; Lisinopril; Metformin; Naproxen; Paracetamol; Pravastatin; Sodium Risedronate (1) Indapamide; Metformin; Amlodipine + Telmisartan (1) Metformin + Vildagliptin; Metformin (1) Lisinopril+Hydrochlorothiazide; Ibuprofen; Ketorolac; Metformin + Vildagliptin; Paracetamol (1) Pravastatin; Metformin; Sodium risedronate; Etoricoxib; Naproxen; Paracetamol; Gliclazide; Lisinopril; Paracetamol; Naproxen (1) Metformin; Empagliflozin (2) Clopidogrel; Metformin; Amoxicillin + Clavulanic acid; Vildagliptin (1)
Renal failure (2)	Ranitidine; Metformin (1) Metformin; Vildagliptin (1)
Acute pancreatitis (1)	Terbinafine; Metformin; Prednisolone; Vildagliptin; Prednisolone; Metformin (1)
Pulmonary hypertension (2)	Indapamide; Metformin; Sertraline; Losartan; Simvastatin (1) Sacubitril + Valsartan; Amlodipine + Valsartan; Rivaroxaban; Bisoprolol; Bisoprolol; Furosemide; Metformin; Budesonide; Digoxin; Spironolactone; Sitagliptin (1)
Dehydration (1)	Dapagliflozin; Furosemide; Metformin (1)
Pancytopenia (1)	Perindopril+Amlodipine; Atorvastatin; Bisoprolol; Clopidogrel; Levothyroxine sodic; Metformin; Rivaroxaban (1)
Drug-induced liver injury (1)	MRNA vaccine against COVID-19 (with modified nucleoside); Metformin (1)

* In suspected drugs, each n corresponds to the drug or drugs suspected of causing the described reaction.

**Table 6 healthcare-11-02197-t006:** Suspected drugs reported in the serious adverse drug reactions belonging to the SOC group “cardiac disorders” that required hospitalization (n being the number of occurrences).

Adverse Drug Reactions Preferred Term (PT) (n)	Drugs (n) *
Heart failure (2)	Metformin (1) Dapagliflozin; Furosemide; Metformin (1)
Palpitations (2)	Metformin + Vildagliptin; Azithromycin; Amoxicillin + Clavulanic acid; Omeprazole; Amlodipine (1) Clopidogrel; Metformin; Nebivolol; Telmisartan + Hydrochlorothiazide; Amlodipine; Amlodipine; Atorvastatin; Furosemide; Pantoprazole (1)
Bradycardia (2)	Glibenclamide + Metformin (1) Lisinopril + Amlodipine; Bisoprolol; Metformin + Vildagliptin; Ramipril (1)
Atrioventricular block, palpitations and bradycardia (1)	Indapamide; Metformin; Sertraline; Losartan; Simvastatin (1)
Atrial fibrillation (1)	Tenecteplase; Metformin (1)
Myocardial infarction (1)	Metformin + Sitagliptin (1)
Tachycardia (2)	Acenocumarol; Folic acid;Omeprazole; Metformin + Sitagliptin; Lisinopril + Hydrochlorothiazide; Methotrexate (1) Methotrexate; Acid folic; Acenocumarol; Metformin + Sitagliptin;Omeprazole; Lisinopril + Hydrochlorothiazide (1)
Brash syndrome (1)	Perindopril + Amlodipine; Bisoprolol; Metformin; Furosemide; Rosuvastatin (1)

* In suspected drugs, each n corresponds to the drug or drugs suspected of causing the described reaction.

**Table 7 healthcare-11-02197-t007:** Metformin and the metformin combination drugs associated with serious reports of reactions of the SOC group “metabolism and nutrition disorders” that resulted in hospitalization (with n being the number of occurrences).

Adverse Drug Reactions-Preferred Term (PT) (n)	Drugs (n) *
Lactic acidosis (25)	Metformin (19) Metformin + Vildagliptin; Metformin (2) Metformin; Sacubitril (1) Sitagliptin; Metformin (2) Metformin + Sitagliptin (1)
Vitamin B_12_ deficiency (4)	Metformin (2) Metformin; Clopidogrel; Perindopril + Amlodipine (1) Perindopril + Amlodipine; Atorvastatin; Bisoprolol; Clopidogrel; Levothyroxine sodic; Metformin; Rivaroxaban (1)
Hypoglycemia (12)	Metformin + Vildagliptin (1) Glibenclamide; Metformin (4) Glibenclamide + Metformin (4) Human insulin; Glibenclamide; Metformin; Acarbose (1) Metformin; Acarbose; Glimepiride (1) Glimepiride; Glibenclamide + Metformin; Human insulin (1)
Decreased appetite (1)	Metformin; Perindopril (1)
Metabolic acidosis (9)	Metformin; Vildagliptin; Gliclazide (1) Metformin; Ibuprofen (1) Metformin (3) Metformin + Sitagliptin (1) Lisinopril + Amlodipine; Bisoprolol; Metformin + Vildagliptin; Ramipril (1) Dapagliflozin; Furosemide; Metformin (1) Clopidogrel; Metformin; Amoxicillin + Clavulanic acid; Vildagliptin (1)
Inadequate control of diabetes mellitus (3)	Metformin; Vildagliptin; Gliclazide (1) Prednisolone; Metformin (1) Metformin (1)
Ketosis (1)	Metformin; Glibenclamide (1)
Dehydration (3)	Metformin (2) Dapagliflozin; Furosemide; Metformin (1)
Hyperlactacidemia (2)	Metformin + Sitagliptin (1) Metformin (1)
Hyperglycemia (1)	Metformin (1)
Metabolic decompensation of diabetes (2)	Metformin; Vildagliptin (1) Dapagliflozin; Furosemide; Metformin (1)
Diabetic ketoacidosis (4)	Dapagliflozin; Furosemide; Metformin (1) Metformin (3)
Euglycemic diabetic ketoacidosis (1)	Clopidogrel; Metformin; Amoxicillin + Clavulanic acid; Vildagliptin (1)

* In suspected drugs, each n corresponds to the drug or drugs suspected of causing the described reaction.

**Table 8 healthcare-11-02197-t008:** Metformin and the metformin combination drugs associated with serious reports of reactions of the SOC “gastrointestinal disorders” that resulted in hospitalization (with n being the number of occurrences).

Adverse Drug Reaction-Preferred Term (PT) (n)	Drugs (n) *
Diarrhea (4)	Metformin + Sitagliptin; Acetylsalicylic acid; Lercanidipine; Olmesartan (1) Metformin (1) Metformin + Vildagliptin (1) Metformin + Sitagliptin (1)
Abdominal pain (3)	Metformin (2) Metformin; Vildagliptin; Gliclazide (1)
Vomiting (9)	Etoricoxib; Gliclazide; Lisinopril; Metformin (1) Metformin (3) Metformin; Vildagliptin; Gliclazide (1) Insulin glargine; Metformin + Sitagliptin; Empagliflozin; Liraglutide (1) Pravastatin; Metformin; Sodium risedronate; Etoricoxib; Naproxen; Paracetamol; Gliclazide; Lisinopril (1) Clopidogrel; Metformin; Amoxicillin + Clavulanic acid; Vildagliptin (1) MRNA vaccine against COVID-19 (with modified nucleoside); Metformin (1)
Oral disorders (1)	Metformin+ Glibenclamide (1)
Epigastric discomfort (1)	Metformin (1)
Nausea (3)	Indapamide; Metformin; Sertraline; Losartan; Simvastatin (1) Clopidogrel, Nebivolol, Metformin; Telmisartan+Hydrochlorothiazide; Amlodipine; Atorvastatin; Furosemide; Pantoprazole (1) Clopidogrel; Metformin; Amoxicillin + Clavulanic acid; Vildagliptin (1)
Hematemesis (1)	Clopidogrel; Nebivolol; Metformin; Telmisartan + Hydrochlorothiazide; Amlodipine; Atorvastatin; Furosemide; Pantoprazole (1)
Duodenal ulcer (1)	Clopidogrel, Nebivolol, Metformin; Telmisartan + Hydrochlorothiazide; Amlodipine; Atorvastatin; Furosemide; Pantoprazole (1)
Sprue-like enteropathy (1)	Metformin + Sitagliptin; Acetylsalicylic acid; Lercanidipine; Olmesartan (1)
Gastrointestinal disorder (1)	Metformin + Sitagliptin; Acetylsalicylic acid; Lercanidipine; Olmesartan (1)
Autoimmune pancreatitis (1)	Metformin and gliclazide (1)
Necrotizing esophagitis (2)	Metformin (2)
Ischemic colitis (2)	Acenocumarol; Folic acid; Omeprazole; Metformin + Sitagliptin; Lisinopril + Hydrochlorothiazide; Methotrexate (1) Methotrexate; Folic acid; Acenocumarol; Metformin + Sitagliptin; Omeprazole; Lisinopril + Hydrochlorothiazide (1)
Retroperitoneal hematoma (2)	Acenocumarol; Folic acid; Omeprazole; Metformin + Sitagliptin; Lisinopril + Hydrochlorothiazide; Methotrexate (1) Methotrexate; Folic acid; Acenocumarol; Metformin + Sitagliptin; Omeprazole; Lisinopril + Hydrochlorothiazide (1)
Mesenteric vein thrombosis (2)	Acenocumarol; Folic acid; Omeprazole; Metformin + Sitagliptin; Lisinopril + Hydrochlorothiazide; Methotrexate (1) Methotrexate; Folic acid; Acenocumarol; Metformin + Sitagliptin; Omeprazole; Lisinopril + Hydrochlorothiazide (1)
Constipation (1)	Metformin + Dapagliflozin; Dapagliflozin (1)

* In suspected drugs, each n corresponds to the drug or drugs suspected of causing the described reaction.

**Table 9 healthcare-11-02197-t009:** Drugs suspected to be involved in adverse drug reactions (ADRs) that result in death (with n being the number of occurrences).

ADR Preferred Term (PT) (n)	Drugs (n) *
Metabolic acidosis (4); Lactic acidosis (5); Renal failure (1); shock (3); Blood pH decrease (1); Respiratory failure (1); Fatigue (1); Neurological Symptoms (1); Toxicity to various agents (1); Renal injury (1); Hyperlactacidemia (1); Acute kidney injury (3); Hyperkalemia (1); Multiple organ dysfunction syndrome (1); Euglycemic diabetic ketoacidosis (2); Aortic thrombosis (1)	Metformin (28)
Hypoglycemic encephalopathy (1)	Metformin; Glibenclamide (1)
Hypotension; Amyloidosis (1)	Chlorthalidone; Metformin; Ramipril (1)
Speech dysfunction (1); Chest pain (1); Hemiparesis (1); Increased blood pressure (1); Nervous system disfunction (1); Cerebrovascular accident (1); Quadriparesis (1); Aphasia(1); Quadriplegia (1)	Gliclazide; Acarbose; Pantoprazole; Alprazolam; Vildagliptin; Atorvastatin; Metformin; Clopidogrel; Ramipril (1)
Euglycemic diabetic ketoacidosis (4); acute kidney injury (2) Metabolic acidosis (2)	Metformin; Empagliflozin (8)
Euglycemic diabetic ketoacidosis (1); Lactic acidosis (1)	Metformin + Dapagliflozin (2)

* In suspected drugs, each n corresponds to the drug or drugs suspected of causing the described reaction.

## Data Availability

The raw data used in this research are available from the authors, depending on INFARMED’s authorization.

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
