# Peer review of "Analysis of Reports Sent to the Portuguese Pharmacovigilance System and Published Literature Regarding the Safety of Metformin in the Elderly"

_healthcare, 2023, doi:10.3390/healthcare11152197_

Round 1

Reviewer 1 Report

It was pleasure reviewing this review on the adverse drug reactions, focusing on the most common and first line drug Metformin used since ages for diabetes mellitus type 2. 

Major comments

The authors have addressed all aspects, but to my opinion the article can be improved. In the current state it seems to be highly chaotic specially when showing the side effects with drugs in combination. 

flow diagram can be used to increase the clarity of the paper

I would suggest the authors to focus on the part where the lit review is compared to the portugese data base. It should be re-written as its not clear at all. 

Why do yu say no relevant articles were found before the year 2010? "After a first search with a longer time 80 frame, no relevant articles were found before 2010, therefore, the search included the pe- 81 riod between January 1st 2010 and September 23th 2022",,, I performed a quick search, and many are shown, based on your established inclusion and exclusion criteria. 

Minor comments

Spelling mistakes and spacing between the words should be corrected.

A proper proof reading would help. 

key words lack the main topic as type 2 diabetes mellitus, which can help improve visibility, if approved.

Spellings and grammatical errors need to be addressed

Author Response

It was pleasure reviewing this review on the adverse drug reactions, focusing on the most common and first line drug Metformin used since ages for diabetes mellitus type 2. 

Author´s response: Thank you very much for your comments and  interest in our work.

Major comments

The authors have addressed all aspects, but to my opinion the article can be improved. In the current state it seems to be highly chaotic specially when showing the side effects with drugs in combination. 

Author´s response: Thank you very much for your comments. The manuscript was now revised according to the comments received, which allowed the quality of this article to improve. Please see the revised manuscript version.

flow diagram can be used to increase the clarity of the paper

Author´s response: Thank you very much for your comments. We added a flow diagram, as suggested, and it made the information transmitted clearer. In addition the tables 4, 5, 6, 7 and 8 were changed and footnotes added. Please see the revised manuscript version.

I would suggest the authors to focus on the part where the lit review is compared to the portugese data base. It should be re-written as its not clear at all. 

Author´s response: Thank you very much for your comments. We changed the text in the suggested way to make it clearer. Please see the revised manuscript version.

Why do yu say no relevant articles were found before the year 2010? "After a first search with a longer time 80 frame, no relevant articles were found before 2010, therefore, the search included the pe- 81 riod between January 1st 2010 and September 23th 2022",,, I performed a quick search, and many are shown, based on your established inclusion and exclusion criteria. 

 Author´s response: Thank you very much for your comments. The search was carried out in the described manner due to the fact that a prior search was performed using a wider period of time and the result was that the articles found weren´t completely suitable for the search we aimed to perform. In fact, the main reason these articles didn’t fit into the goals of our search was that many of these corresponded to studies which sample was made of individuals in which the age was indicated in terms of mean age and these studies didn´t have consistent results relevant to the search performed.  

Minor comments

Spelling mistakes and spacing between the words should be corrected.

A proper proof reading would help. 

Author´s response: Thank you very much for your comments. We corrected these mistakes. Please see the revised manuscript version.

key words lack the main topic as type 2 diabetes mellitus, which can help improve visibility, if approved.

Author´s response: Thank you very much for your comments. We added this key word. Please see the revised manuscript version.

Comments on the Quality of English Language

Spellings and grammatical errors need to be addressed

Author´s response: Thank you very much for your comments. We reviewed the article to address these mistakes. Please see the revised manuscript version.

Reviewer 2 Report

The manuscript "Analysis of Reports Sent to the Portuguese Pharmacovigilance 2 System and Published Literature Regarding the Safety of Met- 3 formin in the Elderly" is interesting and well written. However, I have some main issues I would like the authors to address:

1. According to the methodology part, descriptive analysis was performed, which is also observed in the tables, yet in the discussion part you state that "metformin was safer when used in fixed combination with other  antidiabetics than when used as monotherapy". In my opinion the methodology does not support such a claim.

2.What was the quality of spontaneous reports sent to PPS, did you have any criteria for inclusion or exclusion because of lack of quality?

3. I do not really understand why you combined ADR Reports sent to PPS with literature data. The methodology and the quality of reports may vary substantially. 

4. Conclusion should be revised and only information relevant to the prior parts of the manuscript should remain (ie. nonpharmacological measures were not the aim of the study). 

Author Response

The manuscript "Analysis of Reports Sent to the Portuguese Pharmacovigilance 2 System and Published Literature Regarding the Safety of Met- 3 formin in the Elderly" is interesting and well written. However, I have some main issues I would like the authors to address:

Author´s response: Thank you very much for your comments and the interest you expressed in our work.

  1. According to the methodology part, descriptive analysis was performed, which is also observed in the tables, yet in the discussion part you state that "metformin was safer when used in fixed combination with other  antidiabetics than when used as monotherapy". In my opinion the methodology does not support such a claim.

Author´s response: Thank you very much for your comments. We reviewed and changed this part to improve the quality of the text. Please see the revised manuscript version.

2.What was the quality of spontaneous reports sent to PPS, did you have any criteria for inclusion or exclusion because of lack of quality?

Author´s response: Thank you very much for your comments. In this study, if a report sent to PPS referred to metformin or metformin combined with other medications, then it was included in the analysis.

  1. I do not really understand why you combined ADR Reports sent to PPS with literature data. The methodology and the quality of reports may vary substantially. 

Author´s response: Thank you very much for your comments. We performed the analysis of ADR Reports sent to PPS with literature data search in order to verify whether the information transmitted by these reports was compatible with the documented safety profile and we intend to verify if unknown reactions were being reported Please see the revised manuscript version.

  1. Conclusion should be revised and only information relevant to the prior parts of the manuscript should remain (ie. nonpharmacological measures were not the aim of the study). 

Author´s response: Thank you very much for your comments. We reviewed the manuscript to change the text in the suggested way. Please see the revised manuscript version.

Reviewer 3 Report

As attached

Author Response

Review comments 

This is actually an interesting approach to ensure continuous safety of drugs after launching them into the market. The study is unique as it tries to complement the results of pharmacovigilance with retrospective studies done over a period of time. To make the work better, I have made few suggestions  

Author´s response: Thank you very much for your comments and the interest you expressed when it comes to our work.

Introduction 

Line 45- put “the” before Food….

Line 53-55- Avoid use of single sentence paragraph. 

Line 56- You can paraphrase this sentence as “Before a drug is allowed to be marketed, there is usually a good level of evidence from clinical trials demonstrating efficacy for the indication and population studied, as well as safety regarding the most common adverse reactions”

Line 66- After “For all these reason” put a comma as in “For all these reason,”

Line 69- Use “target” instead of “suspect”. Replace “single or in combination” with “used either as a mono or combination therapy”

Author´s response: Thank you very much for your comments. We reviewed the manuscript according to the excellent suggestions made, except in the suggestion related to suspect drug. In pharmacovigilance, we often use the term suspected drug because in some cases, when the regulatory authority analyzes the reports based on the information they contain, they may conclude that the reported drug did not cause the adverse effect at all, which is why we often use this term. Please see the revised manuscript

Materials and methods 

Line 83-85- You can rewrite this as … the words: “adverse reaction”, “adverse event”, safety” or “pharmacovigilance”; metformin; diabetes; elderly, older people, older patient, older person, geriatric or older adult and Humans (Mesh).  

Lin87-92- This can be rewritten as “In vivo preclinical studies and studies in which drug safety was not described were excluded. Studies with ambiguous design or unclear methods, with non-specific results and those with population younger than 65 years old were also excluded. Finally, studies involving young adults and the elderly whose results were presented as average age and studies where the results were not separated by specific drugs were excluded”.

Line 100- Target drug instead of suspected drug 

Line 107- replace “with” with “who are”

 Author´s response: Thank you very much for your comments. We reviewed the manuscript according to the excellent suggestions made, except in the suggestion related to suspect drug, for the reason described before. Please see the revised manuscript

Results 

Line 187- To increase the comprehensibility of this sentence, add “According to Becquemont et al., after a 3-year…

Line 169-206- I suppose that this subsection is supporting a particular idea. It should be a coherent story that is connected. I suggest you put everything in one or two paragraphs using the right linking words to connect one sentence to the other. The sentences should not just stand alone. 

In Table 1, page 6, the study limitation “Reduced number of patients alone with metformin.” From Lim et al., 2017 Is not clear. Paraphrase it please. No security concerns should be “No safety concerns”

The study population for Schlender at al., 2017 should be “Elderly individuals aged ≥65 with T2DM” Apply same to any where it applies too. 

Line 211- use “was” in place of “has been”

Line 213-215: You can rewrite as “In the analysis of the type of person who reported it, it was concluded that most of the  reports were sent by the MA(write in full) holders (163 reports, which corresponds to 46%), followed by Pharmacists and Physicians with 26% (91 reports) and 24% (85 reports) respectively.

Line 218: “The majority of the ADRs occurred in the range 65 to 74-year-old age group” should be “The majority of the ADRs occurred in the age range 65 to 74years group”

Line 222: Replace “individual with” with “individuals who were”

Line 229: Overall, in SOC group 

Line 231: Table 2

Line 232: According to Table 3

Line 235: Write PT in full here 

Author´s response: Thank you very much for your comments. We reviewed the manuscript according to the excellent suggestions made. Please see the revised manuscript

Table 3: What do you mean by “metformin in combination“ and “metformin and other drugs”? The two groups are combinations. Are they not the same thing?

Please clarify this. 

Author´s response: Thank you very much for your comments. The paragraph before this Table tries to explain this difference: “All the Preferred Term (PT) reactions were analysed and, according to Table 3, PT reactions differ depending on whether metformin is used alone, in a fixed combination, or in combination with other classes of drugs”.

Line 236-237: Using the term most commonly here is not proper. It would make more scientific sense to represent this as percentages of ADR. 

Line 238: Ensure that acronyms are first written in full. Write SmPC in full. 

Line 250-251: Metformin was the target drug most reported. 

Line 252: I would rather use target or reference drugs instead of suspected drugs 

Table 4 should be put in landscape format 

Line 284: Use “Relative to the SOC group, metabolism and nutrition disorders, 

Line 294: I would suggest “14 was life-threatening

Line 302: Table 9

 Author´s response: Thank you very much for your comments. We reviewed the manuscript according to the excellent suggestions made. Please see the revised manuscript

Discussion 

Line 321-323: You cannot draw a conclusion at the beginning of a discussion. This should be moved towards the end. 

Line 332: “Results that meet the known safety profile for these medicines [4]” does not make sense. Please clarify this sentence. 

Line 335: associated with 

Line 346: B12 not B12

 Author´s response: Thank you very much for your comments. We reviewed the manuscript according to the excellent suggestions made. Please see the revised manuscript

Conclusion 

Line 416-417: Rewrite as “Metformin is indicated as first-line treatment for T2DM, but some ADRs have been ascribed to and should be considered in its usage.

Line 418:  metformin-associated ADRs,

Merge the conclusion into one paragraph 

 Author´s response: Thank you very much for your comments. We reviewed the manuscript according to the excellent suggestions made. Please see the revised manuscript

General comments

There are too many abbreviations and acronyms without full meanings. This can easily make the reader lost.  

The use of suspect(ed) drug was quite somehow. I would suggest the use of “target drug” or “drug under study” 

It would be nice to mention or discuss the unique physiological state of the elderly, such as, polypharmacy (use of multiple drugs due to several diseases) due to multiple diseases, organ failures due to age etc. this is important as this unique state affects their response to metformin. 

Author´s response: Thank you very much for your comments. We reviewed the manuscript according to the excellent suggestions made, except in the suggestion related to suspect drug, for the reason described before. Please see the revised manuscript

Reviewer 4 Report

1) The Authors performed an article entitled "Analysis of Reports Sent to the Portuguese Pharmacovigilance System and Published Literature Regarding the Safety of Metformin in the Elderly". Overall, the publication is interesting and is well organized and written.

2) I think “MedDRA” (what is it?; who publishes it; the terminology; etc.) should be better explained in the introduction.

3) “Patients taking metformin extended-release tablets had a lower incidence of adverse effects compared to the treatment with metformin immediate-release tablets”. Please comment this aspect. In my opinion, this information should be discussed in the article. Additionally, are there sustained-release tablets on the market in Portugal?

4) "In general, lactic acidosis, diarrhea, hypoglycemia, and vomiting were the most commonly reported PT reactions (table 3)". A justification/hypothesis for the occurrence of "hypoglycaemia" should be provided according to the mechanism of action of metformin.

5) “In the analysis of the type of person who reported it was concluded that most of the 213 reports were sent by the MA holders (163 reports, which corresponds to 46%), followed by Pharmacists and Physicians with 26% (91 reports) and 24% (85 reports) respectively.” For a more rigorous analysis, I think the number of active pharmacists and physicians in Portugal should be added.

Author Response

1) The Authors performed an article entitled "Analysis of Reports Sent to the Portuguese Pharmacovigilance System and Published Literature Regarding the Safety of Metformin in the Elderly". Overall, the publication is interesting and is well organized and written.

Author´s response: Thank you very much for your comments and the interest you expressed in our work.

2) I think “MedDRA” (what is it?; who publishes it; the terminology; etc.) should be better explained in the introduction.

Author´s response: Thank you very much for your comments. We reviewed the manuscript to explain in a more complete way what “MedDRA” is. Please see the revised manuscript version page 5, line 148.

3) “Patients taking metformin extended-release tablets had a lower incidence of adverse effects compared to the treatment with metformin immediate-release tablets”. Please comment this aspect. In my opinion, this information should be discussed in the article. Additionally, are there sustained-release tablets on the market in Portugal?

Author´s response: Thank you very much for your comments. Unfortunately, the sustained-release tablets aren´t available yet in Portugal

4) "In general, lactic acidosis, diarrhea, hypoglycemia, and vomiting were the most commonly reported PT reactions (table 3)". A justification/hypothesis for the occurrence of "hypoglycaemia" should be provided according to the mechanism of action of metformin.

Author´s response: Thank you very much for your comments. We reviewed the article to clarify this PT reaction according to the mechanism of action of metformin. Please see the revised manuscript version page 6, line 384.

5) “In the analysis of the type of person who reported it was concluded that most of the 213 reports were sent by the MA holders (163 reports, which corresponds to 46%), followed by Pharmacists and Physicians with 26% (91 reports) and 24% (85 reports) respectively.” For a more rigorous analysis, I think the number of active pharmacists and physicians in Portugal should be added.

Author´s response: Thank you very much for your comments.  We agreed with you, unfortunately we do not have access to these numbers so we cannot do that analyse.